# Prediction of Adrenocortical Carcinoma Relapse and Prognosis with a Set of Novel Multigene Panels

**DOI:** 10.3390/cancers14112805

**Published:** 2022-06-05

**Authors:** Xiaozeng Lin, Yan Gu, Yingying Su, Ying Dong, Pierre Major, Anil Kapoor, Damu Tang

**Affiliations:** 1Urological Cancer Center for Research and Innovation (UCCRI), St Joseph’s Hospital, Hamilton, ON L8N 4A6, Canada; linx36@mcmaster.ca (X.L.); guy3@mcmaster.ca (Y.G.); suy36@mcmaster.ca (Y.S.); dongy87@mcmaster.ca (Y.D.); 2The Research Institute of St Joe’s Hamilton, St Joseph’s Hospital, Hamilton, ON L8N 4A6, Canada; 3Department of Surgery, McMaster University, Hamilton, ON L8S 4K1, Canada; 4Department of Oncology, McMaster University, Hamilton, ON L8V 5C2, Canada; majorp@hhsc.ca

**Keywords:** adrenocortical carcinoma, prognostic biomarkers, disease-free survival, overall survival, immune checkpoint proteins, mesenchymal stem cells

## Abstract

**Simple Summary:**

Adrenocortical carcinoma (ACC) is a rare but aggressive cancer with a high rate of fatality. Accurate prediction of cancer relapse following therapy and prognosis (fatality) is essential to improve patient management. This research aims to significantly increase this prediction capacity. We produced four multigene sets: Sig27var25, SigIQvar8, SigCmbnvar5, and SigCmbn_B. These panels have not been studied in ACC and are thus novel. Importantly, they predict ACC’s relapse and death risk with impressively high levels of accuracy. At the disease level, these multigene panels are associated with critical ACC factors, including TP53 gene mutation and changes in immunological processes. Furthermore, we discovered a new ACC factor in predicting ACC relapse and fatality: mesenchymal stem cells (MSCs). Sig27var25, SigIQvar8, SigCmbnvar5, and SigCmbn_B all strongly correlate with MSCs. Collectively, the identification of MSC association with ACC advances our understanding of ACC; Sig27var25, SigIQvar8, SigCmbnvar5, and SigCmbn_B possess significant potential in improving ACC management.

**Abstract:**

Effective assessment of adrenocortical carcinoma (ACC) prognosis is critical in patient management. We report four novel and robust prognostic multigene panels. Sig27var25, SigIQvar8, SigCmbnvar5, and SigCmbn_B predict ACC relapse at area under the curve (AUC) of 0.89, 0.79, 0.78, and 0.80, respectively, and fatality at AUC of 0.91, 0.88, 0.85, and 0.87, respectively. Among their 33 component genes, 31 are novel. They could be differentially expressed in ACCs from normal tissues, tumors with different severity (stages and lymph node metastasis), ACCs with TP53 mutations, and tumors with differentially expressed immune checkpoints (CTLA4, PD1, TGFBR1, and others). All panels correlate with reductions of ACC-associated CD8+ and/or NK cells. Furthermore, we provide the first evidence for the association of mesenchymal stem cells (MSCs) with ACC relapse (*p* = 2 × 10^−6^) and prognosis (*p* = 2 × 10^−8^). Sig27var25, SigIQvar8, SigCmbnvar5, and SigCmbn_B correlate with MSC (spearman r ≥ 0.53, *p* ≤ 1.38 × 10^−5^). Sig27var25 and SigIQvar8 were derived from a prostate cancer (PC) and clear cell renal cell carcinoma (ccRCC) multigene signature, respectively; SigCmbnvar5 and SigCmbn_B are combinations of both panels, revealing close relationships of ACC with PC and ccRCC. The origin of these four panels from PC and ccRCC favors their prognostic potential towards ACC.

## 1. Introduction

Adrenocortical carcinoma (ACC) is an orphan disease with an annual incidence of approximately 0.7–2 cases per million in adults [1,2] and 0.21 cases per million in children [3], which accounts for 0.2% of childhood cancer [4]. The disease affects more women than men with a bimodal age distribution, an early peak in the first decade and a late peak in the fifth and sixth decades of life [2,5,6]. ACC is an aggressive endocrine carcinoma. More than 50% of ACCs produce steroid hormones with clinical consequences; patients with steroid hormone excess have high risks of disease progression and poor prognosis [7]. The estimated 5-year survival rate is less than 50% [8], with median survival around 3–4 years [9]. For patients with organ-confined disease, 5-year survival is 60–80%; for patients with locally advanced ACC or metastatic ACCs, 5-year survival is 35–50% and 0–28%, respectively [10,11]. A recent epidemiological study of 2014 ACC cases in the USA from 1973 to 2014 revealed the disease mortality is 52%, with a median survival time of less than 2 years [5]. Surgical resection is the only curative treatment [12]. However, ACCs show a high relapse rate, with 86% recurrence being reported in 133 ACC patients [13]. Local relapses are commonly associated with metastasis [8], to which therapeutic options are less effective.

Despite being an aggressive carcinoma, ACC has a variable or heterogenous prognosis with either no recurrence or slow metastatic progression in some tumors [9,14]. Effective prediction of ACC prognosis or its clinical behavior at the time of diagnosis is critical for patient management. Clinical outcomes can be estimated by the ACC staging system modified by the European Network for the Study of Adrenal tumors (ENSAT) [8,15]. Other prognosis classifiers include the Ki67 index [8,16], Weiss score [16,17], CpG island methylation, and transcriptome-based classification [8]. CpG island methylator phenotype (CIMP) profile has been used to cluster ACCs into either non-CIMP and high CIMP groups, with the latter being divided into CIMP-high and CIMP-low [18] or three groups consisting of CIMP-high, CIMP-intermediate, and CIMP-low [19]. Increases in CIMP are associated with poor prognosis [18,19]. Based on gene expression profiles, ACCs can be clustered into C1A and C1B, with the former being more aggressive [20]. While CIMP-low ACCs largely belong to the C1B group, both CIMP-intermediate and CIMP-high reside in the C1A group, and CIMP-high show higher overlap with C1A compared to CIMP-intermediate [19]. Thus, both methylation and transcription omics can classify low- and high-risk ACCs, with an overlapping manner.

While both omics can stratify prognostic outcomes, the core events from either omics need to be specified for clinical applications. Towards this goal, hypermethylation of the *G0S2* gene predominantly occurs in CIMP-high ACC and significantly predicts disease-free survival (DFS) and overall survival (OS), with the prediction of DFS showing higher efficiency [21]. With respect to gene expression, the disk large-associated protein 5 (*DLGAP5* or DLG7) and PTEN-induced putative kinase 1 (*PINK1*) genes are predictive of worse DFS, while benzimidazoles 1 homolog beta (*BUB1B*) and *PINK1* expressions are the best predictors of poor OS [20,22].

Even with the above tools, there are no molecular biomarkers in the clinic to evaluate ACC progression and fatality risks [8,16]. The current diagnostic ability in these domains needs to be significantly improved. In our recent effort to investigate prognostic multigene panels for clear cell renal cell carcinoma (ccRCC) and prostate cancer (PC), a 9-gene panel (SigIQGAP1NW) for ccRCC and a 27-gene signature (Sig27gene) for PC were formulated [23,24]. Intriguingly, both signatures robustly predict progression-free survival (PFS, relapse) and OS of ACC at levels exceeding their prognostic potential towards ccRCC or PC, highlighting the credibility of both panels as effective prognostic multigene signatures of ACC. Except for *BIRC5* in Sig27gene, all other component genes in both signatures have not been reported in ACC. From Sig27gene (*n* = 27 component genes) and SigIQGAP1NW (*n* = 9 component genes), we have optimized a signature with 25 (Sig27var25) and 8 (SigIQvar8) variables, respectively, that maintain the prognostic power of Sig27gene and SigIQGAP1NW. Furthermore, from *n* = 6 component genes of Sig27var25 and SigIQvar8, a combined (Cmbn) signature with five variables (SigCmbnvar5) was formulated; SigCmbnvar5 robustly predicts relapse and fatality risk of ACC. In comparison to the most potent published gene panel (*BUB1B* and *PINK1* or BP), Sig27var25, SigIQvar8, and SigCmbnvar5 hold superiority. These panels are associated with the exclusion of immune cells and are highly correlated with the presence of mesenchymal stem cells (MSCs) in ACC. Collectively, we report novel multigene panels that stratify ACC’s relapse and fatality risks with high levels of certainty; these panels have great clinical potential in the management of ACC.

## 2. Materials and Methods

### 2.1. cBioPortal Database

The cBioPortal [25,26] (http://www.cbioportal.org/index.do) (accessed on: 15 March 2022) database contains the most well-organized cancer genetics for various cancer types. The TCGA PanCancer Atlas ACC dataset has *n* = 78 tumors. Tumors have been removed by surgery resection with RNA expression profiled by RNA sequencing (RNA-seq). The suitability of this ACC dataset for overall survival (OS)-related biomarker studies has been demonstrated [27].

### 2.2. Model Size and Variable Optimization

Model size (number of component genes in a multiple gene signature) and variable selection were performed using both the sequential and golden selection (gselection) methods within the BeSS package in R (https://cran.r-project.org/web/packages/BeSS/index.html) (accessed on: 21 March 2022).

### 2.3. Assignment of Signature Scores to Individual ACCs

Signature scores for individual tumors were given using the formula: Sum (coef_1_ × Gene_1exp_ + coef_2_ × Gene_2exp_ + … …+ coef_n_ × Gene_nexp_), where coef_1_ … coef_n_ are the coefs (coefficients) of individual genes, Gene_1exp_ … … Gene_nexp_ are individual gene expressions, and n is the number of component genes, which is *n* = 25 for Sig27var25, *n* = 8 for SigIQvar8, and *n* = 5 for SigCmbnvar5. Individual coefs were derived using multivariate Cox proportional hazards (PH) regression with the R Survival package.

### 2.4. Cutoff Point Estimation

Cutoff points for risk stratifications were determined using the cutpointr (https://github.com/thie1e/cutpointr) (accessed on: 6 April 2022) R package and by Maximally Selected Rank Statistics (the Maxstat package) in R.

### 2.5. Time-Dependent Receiver Operating Characteristic (tROC), ROC, and PR

Analyses of tROC were performed using the R timeROC package. ROC-based area under the curve (AUC) and precision-recall (PR)-AUC were determined with the PRROC package in R.

### 2.6. Examination of Gene Expression

The expression of component genes was determined using the GEPIA2 [28] and UALCAN platform [29].

### 2.7. Correlation Analyses and Heatmap

Correlation (Pearson and Spearman) analyses were performed using tools provided by the cBioPortal, R ggpubr package, and R corrplot packages. Correlations of component gene expression with immune checkpoint protein expression were performed using TISIDB [30]. Oncoprint heatmaps were constructed using the ComplexHeatmap R package [31].

### 2.8. Determination of Immune Cell Infiltration

ACC associated immune cells were profiled using multiple platforms, including CIBERSORT [32], MCP-counter (microenvironment cell populations-counter) [33], Quantiseq [34], xCell [35], and Epic [36] within both immunedeconv [37] and SMDIC R (https://cran.r-project.org/web/packages/SMDIC/index.html) (accessed on: 12 April 2022) packages.

### 2.9. Statistical Analysis

Kaplan–Meier survival analyses and the logrank test were carried out using the R Survival package. Univariate and multivariate Cox regression analyses were run with the R Survival package. A value of *p* < 0.05 is considered statistically significant.

## 3. Results

### 3.1. SigIQGAP1NW and Sig27gene (Sig27) as Potent Prognostic Biomarkers of ACC and Derivation of SigIQvar8 and Sig27var25

We have recently constructed a novel and effective multigene prognostic panel for clear cell renal cell carcinoma (ccRCC) based on the network of IQGAP1 [23]. In our initial screening for its prognostic potential across 33 TCGA cancer types using GEPIA2 [28], SigIQGAP1NW displayed a strong prediction of prostate cancer (PC) relapse or biochemical recurrence (*p* = 9.1 × 10^−5^) and poor OS of ACC (*p* = 1.7 × 10^−4^). The detected prognostic potential in PC is consistent with both PC and ccRCC being urogenital carcinomas. In view of the adrenal gland’s anatomical proximity to the kidney and the blood circulation connections between the two organs, it is intriguing to see a strong prognostic value of SigIQGAP1NW for ACC. The observed prognostic potential of SigIQGAP1NW towards PC would suggest a prognostic value of Sig27gene (Sig27), a prognostic multigene panel of PC constructed using the PC-associated IQGAP1 network [24] for ACC.

To examine the SigIQGAP1NW and Sig27 prognostic value toward ACC, we downloaded the TCGA pan-cancer ACC dataset [27] from cBioPortal. The patient population has typical features of ACC, including the median age of 48.5 and 65% of patients being women (Appendix A), which is consistent with ACC’s peak incidence between 40 and 60 years and 55–60% of patients being women [9]. SigIQGAP1NW and Sig27 scores for individual tumors were generated according to the formula: ∑(coef_i_ × Gene_iexp_)_n_ (coef_i_: Cox coefficient of gene_i_, Gene_iexp_: expression of Gene_i_, *n* = 9 for SigIQGAP1NW and *n* = 27 for Sig27). Cox coefs for individual component genes were generated by multivariate Cox analysis. Both signature scores effectively predict poor OS (Figure 1A) and robustly stratify ACC’s death risk (Figure 1B).

As SigIQGAP1NW and Sig27 were formulated for predicting the fatality risk of ccRCC and recurrence possibility of PC, respectively [23,24], we have determined whether the model or signature size can be optimized. Using both sequential and golden selection (gselection) methods within the BeSS R package, we selected eight variables among the nine component genes of SigIQGAP1NW to formulate SigIQvar8 (Appendix A). For Sig27, the sequential method selected 25 variables (Sig27var25) (Appendix A). Importantly, both SigIQvar8 and Sig27var25 performed slightly better in predicting poor OS of ACC (Figure 1A) and stratifying the fatality risk of ACC (Figure 1C,D). The retainment of 8 of 9 and 25 of 27 component genes in the respective optimized signature validates the prognostic potentials of SigIQGAP1NW and Sig27 for ACC.

We further characterized SigIQvar8 and Sig27var25 scores in the stratification of ACC poor prognosis. In addition to the derivation of cutoff points using Maximally Selected Rank Statistics (see Figure 1B–D), we estimated cutoff points with the empirical, kernel, and normal methods using the cutpointr (https://github.com/thie1e/cutpointr) (accessed on: 6 April 2022) R package. These estimations were performed using bootstrap (*n* = 1000), with an average in-bag sample size being 63.2% of the full sample size. For the estimations of SigIQvar8 score cutoff points, the median ROC-AUC for 1000 in-bag and out-of-bag samples is 0.88 and 0.89, respectively, for the empirical methods and 0.89 (both in-bag and out-of-bag samples) for both kernel and normal methods. Similar performances were also obtained during cutoff point estimation for Sig27var25 scores. The good stratification of fatality risk in out-of-bag samples supports the real-world applications of both signatures. This possibility is in line with the effective stratification of the poor OS using a range of cutoff points determined by the maxstat, empirical, kernel, and normal methods (Figure 1E,F).

We subsequently evaluated the performance of SigIQvar8 and Sig27var25 in discrimination of poor OS using both ROC (receiver-operating characteristic) and PR (precision-recall) curves. SigIQvar8 and Sig27var25 discriminate OS at ROC-AUC 0.88 and 0.91 (Figure 1G) and PR-AUC 0.78 and 0.85, respectively (Figure 1H), revealing both signatures’ ability to effectively stratify ACC fatality risk with Sig27var25 showing slightly better performance. This concept is supported by evaluating both signatures using time-dependent ROC. While the tAUC values were within the 90% range from 18.7 to 71.1 months for SigIQvar8, the tAUC values for Sig27var5 were around the 95% range and above (Figure 1I). Collectively, both SigIQvar8 and Sig27var25 are effective and robust in assessing the fatality risk of ACC.

### 3.2. Sig27var25- and SigIQvar8-Derived Prognostic Potential for PFS

To analyze the prognostic potential for progression-free survival (PFS), we assigned individual tumors with Sig27var25 and SigIQvar8 scores based on an individual gene’s prediction of disease progression using multivariate Cox analysis following the system as described above. Both signatures separate recurrence well, with Sig27var25 showing better performance (Figure 2A,B). Effective stratification of progression risk occurs in a range of cutoff points, which were estimated with 1000 bootstraps (for empirical, kernel, and normal cutoff point estimation), for both signatures (Figure 2C,D). Sig27var25 stratifies ACC progression risk more effectively compared to SigIQvar8 (Figure 2C,D). Sig27var25 score showed superior performance compared to SigIQvar8 in discrimination of ACC progression based on both ROC-AUC and PR-AUC curves (Figure 2E,F).

### 3.3. Characterization of Sig27var25 and SigIQvar8

Both Sig27var25 and SigIQvar8 predict ACC fatality and progression after adjusting for age at diagnosis and tumor stages (Figure 3). We also analyzed individual component genes’ association with ACC progression and poor prognosis. Five variables (LINC01089, RGS11, MXD3, BIRC5, and RAB30) of Sig27var25 are risk factors of poor OS and three of them remain risk factors of fatality after adjusting for age at diagnosis and tumor stages (Table 1). A total of seven component genes, including four of the five genes above, are risk factors for ACC progression even after adjusting for age at diagnosis and tumor stages (Table 1).

For SigIQvar8, two genes (SNHG10 and RECQL4) are risk factors for ACC progression after adjusting for age at diagnosis and tumor stage (Table 2). Long non-coding RNA (lncRNA) LINC01089, lncRNA LOC100128288 are risk factors for poor OS and shortening PFS with no adjustment for age at diagnosis and tumor stage (Table 2). LINC01089 is a common component gene in Sig27var25 and SigIQvar8 (Table 1 and Table 2).

### 3.4. Construction of SigCmbnvar5

Sig27var25 and SigIQvar8 are highly effective at predicting ACC prognosis and progression; both signatures contain component genes predicting poor OS independently of age and tumor stages, for instance SNHG10 and RECQL4 for SigIQvar8 (Table 2) as well as MXD3, BIRC5, and RAB30 for Sig27var25 (Table 1). We noticed the significant small *p*-values of MXD3, BIRC5, RAB30, SNHG10, and RECQL4 in predicting poor OS and worse PFS (Table 1 and Table 2). Despite the associated HRs being unimpressive when their continuous gene expression data were used (Table 1 and Table 2), the low *p*-values suggest the variables’ potential in risk stratification. Indeed, MXD3, BIRC5, RECQL4, and SNHC10 robustly separate ACCs with high risks of fatality and progression, while RAB30 effectively stratifies progression risk (Appendix A). The HRs for fatality and progression in the stratified group range from 3.2 to 14.3, *p* < 0.01 (Appendix A). As lncRNA LINC01089 is present in both Sig27var25 and SigIQvar8 (Table 1 and Table 2), we thus analyzed the signature potential of the six combination (Cmbn) genes (LINC01089, MXD3, BIRC5, RAB30, RECQL4, and SNHC10) of both Sig27var25 and SigIQvar8. When modeling their prediction of OS using the Cox model within the BeSS R package, both sequential and gsection methods selected five variables (SigCmbnvar5) among these six genes with MXD3 being excluded (Appendix A). The elimination of MXD3 from SigCmbnvar5 is supported by its high levels of correlation with SNHG10, RECQL4, and BIRC5 (Appendix A).

SigCmbnvar5 predicts ACC fatality and progression in comparable effectiveness to Sig27var25 and SigIQvar8 (comparing Figure 4A to Figure 3) and stratifies the fatality risk in a range of cutoff points at 85% sensitivity and 84% specificity (Figure 4B); the discriminative value of poor OS is at ROC-AUC 0.85 (Figure 4D) and PR-AUC 0.77 (Figure 4E). SigCmbnvar5 effectively stratifies poor PFS (Figure 4C–E). Collectively, SigCmbnvar5 predicts the risk of ACC fatality and progression with high degrees of certainty.

### 3.5. Superiority of Our Prognostic Multigene Signatures to Those Previously Reported

There are a limited number of reports on ACC prognostic biomarkers. Based on microarray and reverse qPCR analyses, the gene pair DLG7 and PINK1 was the best predictor of ACC progression, and the gene pair of BUB1B and PINK1 displayed the best prediction of ACC poor prognosis [20,22]. We first evaluated both gene pairs for their predictive values toward OS and PFS. Using our system, we confirmed both pairs to significantly predict ACC progression and fatality (Appendix A); nonetheless, BUB1B-PINK1 (BP) and DLGAP5 (DLG7)-PINK1 (DP) predict poor OS and worse PFS with comparable efficiency (Appendix A). Based on tROC, ROC, and PR curves, BP showed slightly better stratification for both worse PFS and poor OS (Appendix A). We further noticed an exceptionally high level of correlation between BUB1B and DLGAP5 expressions with Spearman r = 0.92 and *p* = 1.61 × 10^−31^ (Appendix A), consistent with their function in mitotic spindle organization [38,39,40]. Collectively, evidence supports that both BP and DP have a high level of similarity and BP is a slightly better prognostic panel of ACC.

Both Sibcmbnvar5 and the published BP signature (Sigpub_BP) display good prognostic values (Figure 4 and Appendix A) and are composed of a small number of component genes. We thus analyzed the six combination genes (LINC01089, MXD3, BIRC5, RAB30, RECQL4, and SNHC10) of Sig27var25 and SigIQvar8 as well as two component genes (BUB1B and PINK1) of Sigpub_BP for their potential to form a multigene panel. The BeSS-based model selected a panel of six genes with the exclusion of BIRC5 and PINK1 (Appendix A); their elimination is largely attributable to their high levels of correlation with other genes (Appendix A), i.e., their presence does not add to the biomarker potential of the resultant multigene panel (SigCmbn_B) in predicting ACC relapse and prognosis. SigCmbn_B, thus, consists of LINC01089, RAB30, SNHC10, MXD3, RECQL4, and BUB1B (Appendix A). The panel robustly stratifies poor OS and worse PFS (Appendix A).

We then compared the four multiple gene panels (Sig27var25, SigIQvar8, and SigCmbnvar5, and SigCmbn_B) with the published BP signature: Sigpub_BP. In reference to 95% CI and the respective *p*-values, Sig27var25, SigIQvar8, SigCmbnvar5, and SigCmbn_B exhibit better predictions of OS and PFS compared to Sigpub_BP (Appendix A). With empirically estimated cutoff points, Sig27var25, SigIQvar8, SigCmbnvar5, SigCmbn_B stratify the fatality and progression risk better than Sigpub_BP, as evident by the survival curves, sensitivities, and specificities (Figure 5A,B). The superiority of Sig27var25, SigIQvar8, SigCmbnvar5, and SigCmbn_B to Sigpub_BP is also supported by their ROC and PR curves of OS and PFS (Figure 5C,D). Nonetheless, these five gene panels are related, as evident by their high correlations (Appendix A), with all signatures identifying events (poor prognosis and progression) in a partially overlapping manner (Appendix A). SigCmbn_B correlates more to SigCmbnvar5 (r = 0.98) than to Sigpub_BP (r = 0.77) (Appendix A), supporting both SigCmbnvar5 and SigCmbn_B being better multigene panels compared to Sigpub_BP. Collectively, Sig27var25, SigIQvar8, SigCmbnvar5, and SibCmbn_B add significant clinical values in assessing ACC recurrence and prognosis.

### 3.6. SigIQvar8, Sig27var25, SigCmbnvar5, and SigCmbn_B Being Novel to ACC

SigIQvar8 consists of eight component genes, including long non-coding RNA (lncRNA) LINC01089, lncRNA LOC100128288, AI894139 pseudogene LOC155060, hect domain and RLD 2 pseudogene 2 HERC2P2, a non-protein coding RNA SNHG10 and three protein-coding genes (RECQL4, ATXN7L2, and THSD7A) (Table 3). LncRNAs are known to regulate gene expression in part via their sponge actions towards miRNAs [41,42], and miRNAs commonly regulate multiple targets [43].

LINC01089 has been reported to negatively impact tumorigenesis and play a role in the inhibition of the Wnt/β-catenin signaling (Table 3) [44,45]; this impact might be in part via the targeting of miRNAs, including miR-3187-3p [46] and miR-27a [47]. β-catenin is a driver gene in aggressive ACC subgroups [19]. The positive correlation of LINC01089 with poor OS (HR 1.002, *p* < 0.05; Table 1 and Table 2) indicates that LINC01089 likely does not suppress β-catenin in ACC. However, LINC01089 also promotes gastric cancer via sponging miR-145-5p [48] and resistance to sorafenib in hepatocellular carcinoma cells by targeting miR-665 [49]. SNHG10 enhances hepatocarcinogenesis and metastasis (Table 3) [50]. HERC2P2 was recently identified as a component gene in a 10-gene panel of blood transcripts that classifies the risk of breast cancer (Table 3) [51]. RECQL4 is one of five human RecQ helicases, with others being RecQ1, WRN, BLM, and RecQ5. Mutations in WRN, BLM, and RECQL4 cause Werner syndrome (WS), Bloom syndrome (BS), and Rothmund–Thomson syndrome (RTS), respectively, which are associated with premature aging, cancer predisposition, and chromosome abnormalities [52]. Elevations in RECQL4 display oncogenic activities in prostate cancer [53] and promote chemoresistance in gastric cancer (Table 3) [54]. The functionality of RECQL4 in regulating genome stability [52] is in line with abnormalities in DNA damage response being detected in ACC [21] and the typical chromosomal aneuploidy observed in ACC [55]. Collectively, evidence supports the important roles of RECQL4 in promoting tumorigenesis. Importantly, none of these genes have been reported in ACC (Table 3).

Among the 25 component genes of Sig27var25, 18 genes show activities relevant to oncogenesis and 7 genes with unknown tumorigenic roles (Table 4). VGF facilitates resistance to tyrosine kinase inhibitors in lung cancer [56]. RGS11 is a biomarker of lung cancer [57]. Evidence supports MXD3 in the promotion of medulloblastoma [58] (Table 4). BIRC5, or Survivin, is a well-studied anti-apoptotic protein promoting tumorigenesis and progression [59,60]. LTC4S is a component gene in an immune signature associated with clinical response in breast cancer [61]. Evidence supports negative impacts on tumorigenesis for FPR3 [62], RAB30 [63], RIPOR2 (FAM65B) [64], PLXNA4 [65,66], MCTP1 [67], and KCNN3 [68] (Table 4).

NOD2 was implicated in the immunosuppression of gastric cancer [69]. Blood PI15 is a biomarker of cholangiocarcinoma [70]. LAMP3 is associated with aggressive breast cancer [71]. Increases in HDAC9 were observed in basal bladder cancer [72]. ZFHX4 is one of the nine under-expressed genes and is a susceptibility locus of cutaneous basal cell carcinoma [73]. TFEC regulates mTOR activation via lysosome biogenesis [74] (Table 4).

Except for *BIRC5*, the remaining 24 genes are novel to ACC (Table 4). Upregulation of BIRC5 was reported in ACC compared to benign adrenocortical adenomas and normal adrenal glands, and the upregulation may associate with poor ACC prognosis (*p* = 0.053) [75]. Collectively, Sig27var25 is a novel prognostic panel for ACC.

Among its five component genes (LINC01089, SNHG10, RECQL4, BIRC5, and RAB30), only BIRC5 is known for contributing to ACC [75], and thus, Sigcmbnvar5 is a novel prognostic signature of ACC. In the same logic, SigCmbn_B is novel, evident by five of its six component genes (LINC01089, RAB30, SNHC10, MXD3, RECQL4, and BUB1B) being novel.

### 3.7. Differential Expression of Specific Component Genes during ACC Pathogenesis

The novelty of Sig27var25, SigIQvar8, and SigCmbnvar5, as discussed above, is relevant to ACC, which is supported by differential expressions of their component genes following ACC tumorigenesis. Downregulations of three lncRNAs, LINC01089, SNHG10, and HERC2P2, in ACCs were observed compared to normal adrenal gland tissues (Figure 6). A significant reduction in LINC01089 in CIMP-high and CIMP-low ACC (Figure 6) is in line with its observed negative impact on the Wnt/β-catenin [44,45]. On the other hand, RECQL4 is significantly upregulated in CIMP-high and CIMP-intermediate ACCs (Figure 6), two aggressive clusters of ACC [19]. Downregulations of Sig27var25 component genes (FPR3, LCN12, RAD30, RGS11, TFEC, and VGF) occur in either CIMP-high, CIMP-intermediate, or both (Figure 6). Along with confirmation of BIRC5 upregulation in CIMP-high and CIMP-intermediate ACC, HAGHL, MXD3, and PRR7 are also upregulated in either CIMP-high, CIMP-intermediate or both (Figure 6). The alterations of these component genes in aggressive CIMP-high and CIMP-intermediate ACCs support the potent biomarker potential of Sig27var25, SigIQvar8, and SigCmbnvar5 in the prediction of ACC poor prognosis and progression.

Additionally, high levels of RECQL4, MXD3, and PRR7 mRNA expression are associated with advanced tumor stages (Figure 7A). Significant elevations of PRR7 mRNA expression correlate with lymph node metastasis (Figure 7B). Intriguingly, elevations of RECQL4, MXD3, BIRC5, and SNHG10 expression are associated with TP53 mutations (Figure 7C). Considering TP53 mutations as a major genomic alteration in ACC [76,77], these genes likely cooperate with TP53 mutations to facilitate ACC tumorigenesis. These observations suggest a mechanistic correlation of Sig27var25, SigIQvar8, and SigCmbnvar5 with a key oncogenic process, i.e., TP53 mutations, of ACC.

### 3.8. Association of Multigene Panels with Immunosuppressive ACC Microenvironment

TP53 mutation is known for its correlations with the immunosuppressive microenvironment [78,79]. The above observations, thus, indicate correlations with immunosuppressive phenotype as a potential mechanistic association for our multigene panels. In line with this inference, we observed a common correlation of BIRC5, MXD3, and RECQL4 with TGFBR1 (Figure 8). Together with the observed correlation of RGS11 with TGFB1 (Figure 8), evidence supports elevations in the TGFβ signaling in ACCs with increases in Sig27var25, SigIQvar8, and SigCmbnvar8 scores. TGFβ signaling plays a major role in shaping the immunosuppressive tumor microenvironment [80,81]. Through transduction of IL10-induced signaling, IL10RB (IL10 receptor subunit beta) contributes to the immunosuppressive tumor microenvironment [82,83]. Consistent with this knowledge, BIRC5, MXD3, and RECQL4 expressions positively correlate with IL10RB expression (Figure 8). NOD2 expression positively correlates with the expression of a set of co-inhibitory receptor or immune checkpoint receptors CTLA4, LAG3, PD-1 (PDCD1), and TIGIT (Figure 8). These co-inhibitory receptors play an essential role in immune tolerance and the formation of an immune permissive microenvironment [84]. The checkpoint inhibitor TIGIT (T cell immunoreceptor with immunoglobulin and ITIM domain) plays a critical role in immune evasion of cancers via binding CD112 (PVRL2 or nectin-2), one of two TIGIT ligands [85]. Intriguingly, VGF expression significantly correlates with PVRL2 expression (Figure 8). As those component genes (Figure 8) are within Sig27var25, SigIQvar8, and SigCmbnvar5, evidence thus supports a correlation of these signatures with immune escape in ACC.

The major goal of immune evasion is to avoid CD8+ T cells and NK cell-mediated killing of tumor cells. In line with this knowledge, Sig27var25 scores are significantly correlated with reductions of CD8+ T cells and NK cells in ACC, determined by multiple computation programs, including xCell, ssGSEA (Figure 9A,B), GIBERSORT, ssGSEA, Epic, and MCP (Appendix A). Reverse correlations of Sig27var25 scores with both cytotoxic cells (Figure 9C) and cytotoxic scores (Appendix A) were also observed. Correlations of SigCmbnvar5 and SigCmbn_B with NK cells, CD8+ cells, and cytotoxic cells, correlation of SigIQvar8 with NK cells, as well as correlations of Sigpub_BP with NK cells and cytotoxic cells were also detected (Figure 9D). Collectively, the above results clearly reveal an association of Sig27var25, SigIQvar8, SigCmbnvar5, and SigCmbn_B with the immunosuppressive environment of ACC.

### 3.9. Correlation of Multigene Signatures with Mesenchymal Stem Cells (MSCs)

MSCs migrate to tumor mass and contribute to cancer progression [86]. Nonetheless, the association of MSCs with ACC remains unknown. By taking advantage of profiling the MSC presence in ACC using the xCell program within the MSDIC R package, we noticed the MSC content showed predictive power for ACC fatality (HR 301.5, 95% CI 27.89–3235, *p* = 2.59 × 10^−6^) and progression (HR 249.2, 95% CI 24.77–2507, *p* = 2.81 × 10^−6^). ACCs with high levels of MSCs are significantly associated with poor OS and rapid disease progression (Figure 10A). Interestingly, these ACCs also have increased Sig27var25 scores (Figure 10B). MSC content in ACC also significantly correlates with SigIQvar8, SigCmbnvar5, SigCmbn_B, and Sigpub_BP (Figure 10C). The observed prognostic potential of MSCs in ACC is novel; the detected correlations of Sig27var25, SigIQvar8, SigCmbnvar5, SigCmbn_B, and Sigpub_BP with MSCs provide additional support for their prognostic potentials.

## 4. Discussion

Effective assessment of prognosis and prediction of cancer recurrence is essential in the management of patients. This is particularly critical for ACC owing to its rarity, aggressiveness, and heterogeneous prognosis [16]. However, this clinical capacity remains generally poor. ACC prognosis can be estimated with good accuracy by using (1) CIMP (CpG island methylation phenotype) clustering [18,19,87], (2) gene expression profiles-based clustering (C1A and C1B) [20], (3) COMBI score (combination of modified GRADE (Grading of Recommendations Assessment, Development and Evaluation) score and genomic alterations) [88], and (4) BUB1B-PINK1-based gene expression [20,22]. Nonetheless, there is no molecular knowledge or biomarkers in clinical risk assessment being recommended by the European Society of Endocrinology Clinical Practice Guidelines on ACC [9]. The current knowledge and capacity in risk (progression and fatality) prediction remains insufficient. The set of multigene prognostic panels reported here fulfills this gap.

The unique feature of this study is the derivation of ACC prognostic biomarkers from the relevant knowledge of ccRCC and PC, two urogenital carcinomas. According to the molecular knowledge obtained from pan-cancer studies, the CoC (cluster of cluster) I ACCs, which have a good prognosis, share lower rank proliferative features with KIRC (ccRCC) and PC [55]. This similarity may not underline the similar prognostic properties of ccRCC and PC biomarkers for ACC. It is particularly intriguing considering PC’s generally nonaggressive nature, Sig27′s predictive power for PC recurrence, and Sig27var25 being the most robust multigene panel in predicting ACC prognosis and progression. The physical location of the adrenal gland to the kidney might be partially attributable to the shared prognosis features between ACC and ccRCC. The central role of the adrenal gland in hormone synthesis [89] might underline the relationship of PC Sig27 to ACC prognosis. In this regard, it might be worthwhile exploring the clinical applicability of the standard of care in ccRCC and PC in ACC therapy. For instance, will sunitinib (commonly used in treating metastatic ccRCC) have clinical benefits toward advanced ACC? Will the second-generation anti-androgens, abiraterone and enzalutamide, be effective in treating ACCs producing adrenal androgens, the second most common hormone produced by ACC [16]?

Importantly, both the ccRCC and PC multigene panels, SigIQvar8 and Sig27var25, are highly effective in assessing ACC prognosis and progression. Both panels together with their combination signature, SigCmbnvar5, outperform one of the most effective ACC prognostic panels, BUB1B-PINK1. Sigpub_BP was derived from an elegant microarray analysis, including *n* = 34 ACCs [20]; these tumors were clustered into C1A and C1B subgroups based on the expression profile of 746 prob sets [20]. ACCs in CIA (*n* = 23) and CIB (*n* =13) are associated with high and low risks of progression (relapse/metastasis) and prognosis (deaths), respectively [20]. This clustering system stratifies the relapse risk at 89.5% sensitivity and 73.3% specificity, and prognosis at 94.1% sensitivity and 70.6% specific [20]. Our panel Sig27var25 predicts ACC relapse at a sensitivity of 80% and specificity of 92.1%, and prognosis at a sensitivity of 85.2% and specificity of 88.2% (Figure 5). Sig27var25 stratifies relapse and prognosis risk with more balanced sensitivity and specificity compared to C1A-C1B. Additionally, C1A-C1B consists of 746 genes, and Sig27var25 contains 25 component genes. Evidence, thus, supports Sig27var25 at least matches the performance of C1A-C1B in predicting ACC progression and prognosis and adds significant value to the current ACC risk assessment. Sigpub_BP has been retrospectively validated by two groups [21,22]. The high levels of correlation (spearman r ≥ 0.61) between Sig27var25, SigIQvar8, or SigCmbnvar5 and Sigpub_BP (Appendix A) and their partially overlapping manner in prognosis and progression stratifications (Appendix A) provide indirect validation of our multigene panels. Additionally, all signatures performed exceptionally well in out-of-bag samples (*n* = 1000) when cutoff points were estimated (see 3.1, paragraph 4). The combination of knowledge generated in this study with knowledge previously published resulted in SigCmbn_B, which performs equivalently or marginally better then SigCombnvar5 (Figure 5). The availability of SigCmbnvar5 and SigCmbn_B offers options and cross validation in clinical applications. For instance, situations may arise in which either or both panels can be assessed.

While RNA-sequencing allows the detection of multiple genes to be highly feasible, multigene panels with a small number of component genes might still offer unique advantages. This feature is particularly relevant to ACC. Considering its rarity and the need for an expert team in patient care, establishing a network of risk assessment will improve decision making or personalized medicine. This might require an initial assessment from small and not well-equipped centers using classic methodologies such as quantitative real-time PCR to analyze a limited number of genes. SigCmbnvar5, SigCmbn-B, and SigIQvar8 are the choice for initial risk (both progression and prognosis) assessment. This initial assessment can be further simplified if necessary. For instance, RAB30 and RECQL4 predict relapse and prognosis risk at HR 5.5 (*p* = 8.21 × 10^−6^) and HR 12.7 (*p* = 4.38 × 10^−7^) (Appendix A) respectively; both genes can be used for initial risk assessment. More thorough evaluations can then be followed with Sig27var25. A range of cutoff points, which are defined in this study (Figure 2C,D), are the starting points to stratify progression and prognosis risk. Nonetheless, Sig27var25, SigIQvar8, SigCmbnvar5, and SigCmbn_B should be validated both retrospectively and prospectively, along with polishing cutoff points in the future. Furthermore, there exist needs to integrate our multigene panels with the current knowledge (Weiss score, Ki67 index, C1A-C1B, CoC, CIMP, and COMBI score) to cover different aspects relevant to ACC progression and prognosis to better serve the clinical needs.

Future validation is supported by the novelty of our multigene panels. Except for BIRC5, all component genes are novel to ACC (Table 3 and Table 4). Particularly appealing is the high stratification potential of SNHG10, RECQL4, MXD3, BIRC5, and RAB30 towards prognosis and progression as individual genes (Appendix A). RECQL4 is particularly robust in assessing poor prognosis and RAB30 is highly effective in predicting rapid progression (Appendix A). The observed differential expressions of signature component genes in ACC and tumors with adverse features (high stage and lymph node metastasis) support the relevance of these panels as valuable prognostic biomarkers of ACC.

The effectiveness of Sig27var25, SigIQvar8, SigCmbnvar5, and SigCmbn_B might be attributable to their association with important oncogenic processes of ACC. RECQL4, MXD3, BIRC5, and SNHG10 are significantly associated with TP53 mutations, a key molecular feature of ACC [16,55]. RECQL4′s role in DNA repair [52] may in part contribute to its association with TP53 mutations in ACC. The underlying mechanisms for the associations of MXD3, BIRC5, and SNHG10 with TP53 mutations require further investigations. Despite these uncertainties, their relationship with TP53 might contribute to the biomarker potential of the multigene panels in which they are involved. Consistent with this possibility, RECQL4, MXD3, BIRC5, and SNHG10 are presented with different combinations in all multigene panels. For instance, MXD3 and BIRC5 are in Sig27var25 (Table 1); RECQL4 and SNHG10 are component genes of SigIQvar8 (Table 2); SNHG10, RECQL4, and BIRC5 are major component genes of SigCmbnvar5 (see Section 3.4); and MXD3, SNHG10, and RECQL4 are included in SigCmbn_B (see Section 3.5). TP53 mutation is an established risk factor for ACC; the mutation was observed in 50–80% of childhood ACCs [90,91,92] and approximately 10% of adult ACC [19]. TP53 is one of the driver genes in CoC (cluster of cluster) II and III ACCs, which are C1A ACCs and have a high risk of tumor progression [19,20]. Evidence, thus, supports the associations of RECQL4, MXD3, BIRC5, and SNHG10 with TP53 mutant (Figure 7C) as a contributing factor to their potential in predicting ACC progression and prognosis. However, the biomarker potentials of these four genes and the individual multigene panels in predicting ACC relapse and prognosis are likely attributable to multiple factors in addition to TP53 mutations. The same scenario may also apply to the association of individual multigene panels with an immunosuppressive microenvironment of ACC. Nonetheless, it will be interesting to investigate the contributions of TP53 association to multigene panels’ biomarker potential and their correlation with ACC evasion of immune attacks.

A typical mechanism for ACC to evade immune reactions is to exclude immune cells [55]. Consistent with this knowledge, Sig25var25 component genes associate with TGFβ signaling (Figure 8), a critical contributor to T-cell exclusion [81]; Sig27var25, SigIQvar8, SigCmbnvar5, SigCmbn_B, as well as Sigpub_BP significantly associate with reductions of CD8+ T and/or NK cells in ACC (Figure 9). In view of hypercortisolism being the most common clinical feature in ACC patients with hormone excess [16] and the well-established anti-inflammatory roles of corticosteroids, there is a clear need to select the right ACC patients for immunotherapy [93]. In this regard, it will be interesting to investigate the potential association of multigene panels reported here with cortisol levels in ACC patients. This knowledge will be useful in improving immunotherapy in ACC patients.

A novel association is between these signatures with MSCs (Figure 10), a key player that promotes cancer progression [86] with an important function in shaping the immunosuppressive microenvironment [94]. Nonetheless, MSC’s role in ACC remains unknown. We demonstrated a significant stratification potential of ACC progression and poor prognosis by MSCs, and high levels of correlation of all signatures, including Sigpub_PB with MSC (Figure 10). This knowledge is not only novel but also highly relevant to ACC.

The multigene panels reported here are based on the gene expression profile. In view of the critical contributions of other omics to ACC progression, including epigenetics (non-coding RNA and DNA methylation), genomic alterations (mutations and copy number changes), and protein profile alterations, the relationship of our panels to these omics can be explored in the future; this will not only advance our knowledge on ACC but also improve the assessment of relapse and prognosis. In this regard, the three omics (gene expression, copy number alterations, and mRNA profile) have been integrated according to the knowledge of Nottingham Prognostic Index (NPI) of breast cancer [95]. Modeling of the gene similarity network (GSN) of the three omics, which were produced by t-distributed stochastic neighbor embedding (t-SNE), using the visual geometry group at the 33 layer (VGG-33) and residual neural network consisting of 112 layers (ResNet-112) can predict breast cancer OS following surgery with an AUC of 0.9999 (VGG-33) and 0.9991 (ResNet-112) [95]. In principle, a similar approach can be explored for our multigene panel-produced risk scores. This potential research direction is further supported by NOD2 being a component of the genes included in the above breast cancer models [95] as well as in our panel Sig27var25 (Table 4). Additionally, NOD 2 independently predicts ACC relapse after adjusting for age at diagnosis and tumor stage (Table 1).

This work is not without limitations. The TCGA pan-cancer ACC population contains only primary tumors, and more than 50% of these cases are stage I and stage II tumors (Appendix A). This may not reflect the current clinic situation in which most tumors are diagnosed as advanced tumors at stage III and stage IV [12,16]. However, this situation will likely change as early diagnoses become more prevalent in the future. Nonetheless, the robustness of the panels is evident as Sig27var25, SigIQvar8, and SigCmbnvar5 (as well as SigCmbn_B, data not shown) and even some component genes predict ACC fatality and progression independently of tumor stages (Figure 3 and Table 1 and Table 2).

## 5. Conclusions

We report a set of novel and highly robust multigene panels for prediction and stratification of ACC recurrence and poor prognosis. These panels were not directly formulated based on ACC features but on the prognosis of ccRCC and recurrence of PC. This indirect approach significantly validates the clinical potential of Sig27var25, SigIQvar8, SigCmbnvar5, and SigCmbn_B. These panels correlate with the key processes of ACC, including TP53 mutation and lymphocyte exclusion. We detected a novel prognostic feature of MSCs in ACC. In view of the demonstrated importance of MSCs in cancer progression, the observed prognostic values of MSCs in ACC are likely highly relevant. Furthermore, our multigene panels display high correlations with ACC-associated MSCs. Our panels outperform the available or published ACC prognostic biomarkers. The combination of Sig27var25, SigIQvar8, SigCmbnvar5, and SigCmbn_B, along with published signatures, including Sigpub_BP, will likely advance the current clinical capacity in risk prediction and stratification.

## 6. Patents

This research has resulted in a USA provisional patent application.

## Figures and Tables

**Figure 1 cancers-14-02805-f001:**
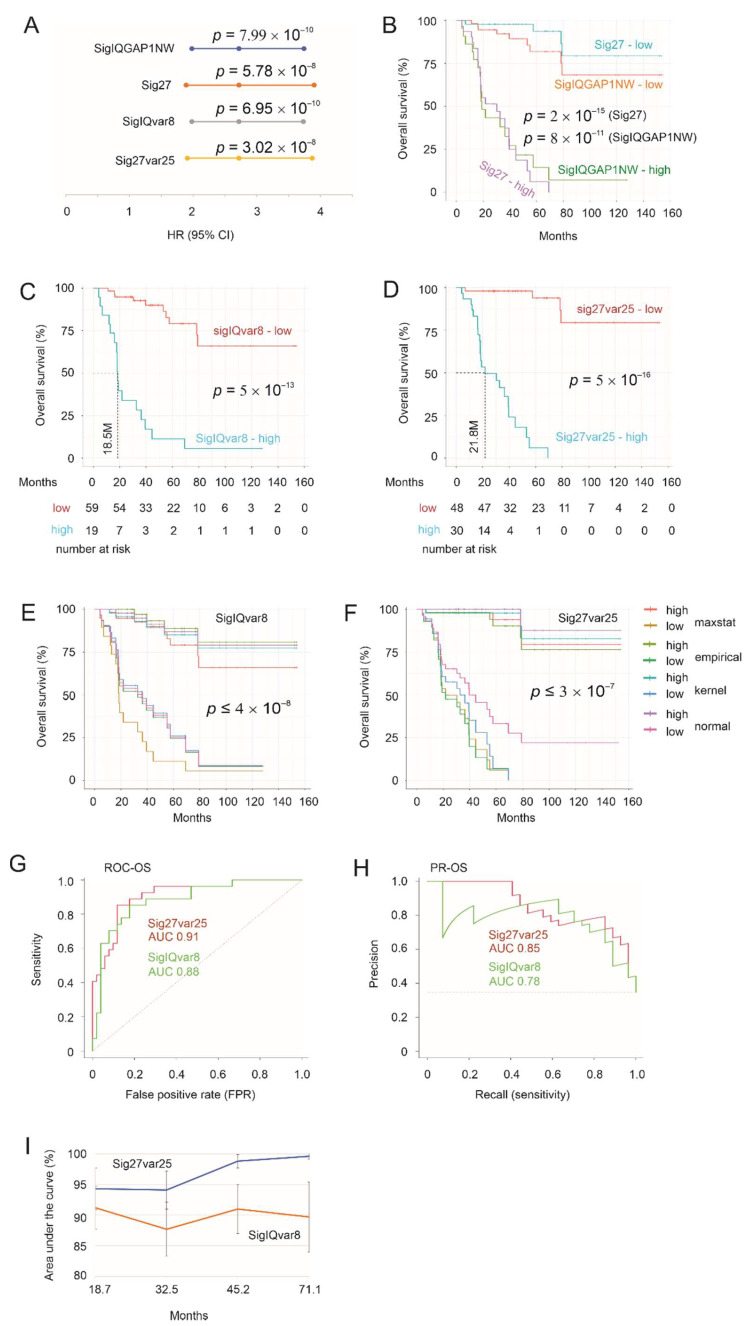
Effective prediction of poor OS of ACC by Sig27var25 and SigIQvar8. (**A**) Hazard ratio (HR), 95% confidence interval (CI), and *p* values for the indicated multigene signatures in predicting poor OS. (**B**) Stratification of fatality risk by Sig27 and SigIQGAP1NW. Cutoff points were estimated by Maximally Selected Rank Statistics. Kaplan–Meier curves were constructed using the R survival package. Statistical analyses were performed using logrank test. (**C**,**D**) Separation of ACCs with high fatality risk from those with low risk by SigIQvar8 (**C**) and Sig27var25 (**D**). Cutoff points were estimated by Maximally Selected Rank Statistics. (**E**,**F**) Stratification of fatality risk with the indicated gene panels using cutoff points estimated by maxstat (Maximally Selected Rank Statistics), empirical, kernel, and normal methods using the R cutpointr package. (**G**,**H**) ROC and PR curves to evaluate the performance of Sig27var25 and SigIQvar8 in fatality risk stratification. The curves were produced using the PRROC package in R. (**I**) Time-dependent ROC-AUC for the indicated multigene panels. Error bands are for standard error (SE). Time-dependent ROC-AUC values were obtained using the R timeROC package.

**Figure 2 cancers-14-02805-f002:**
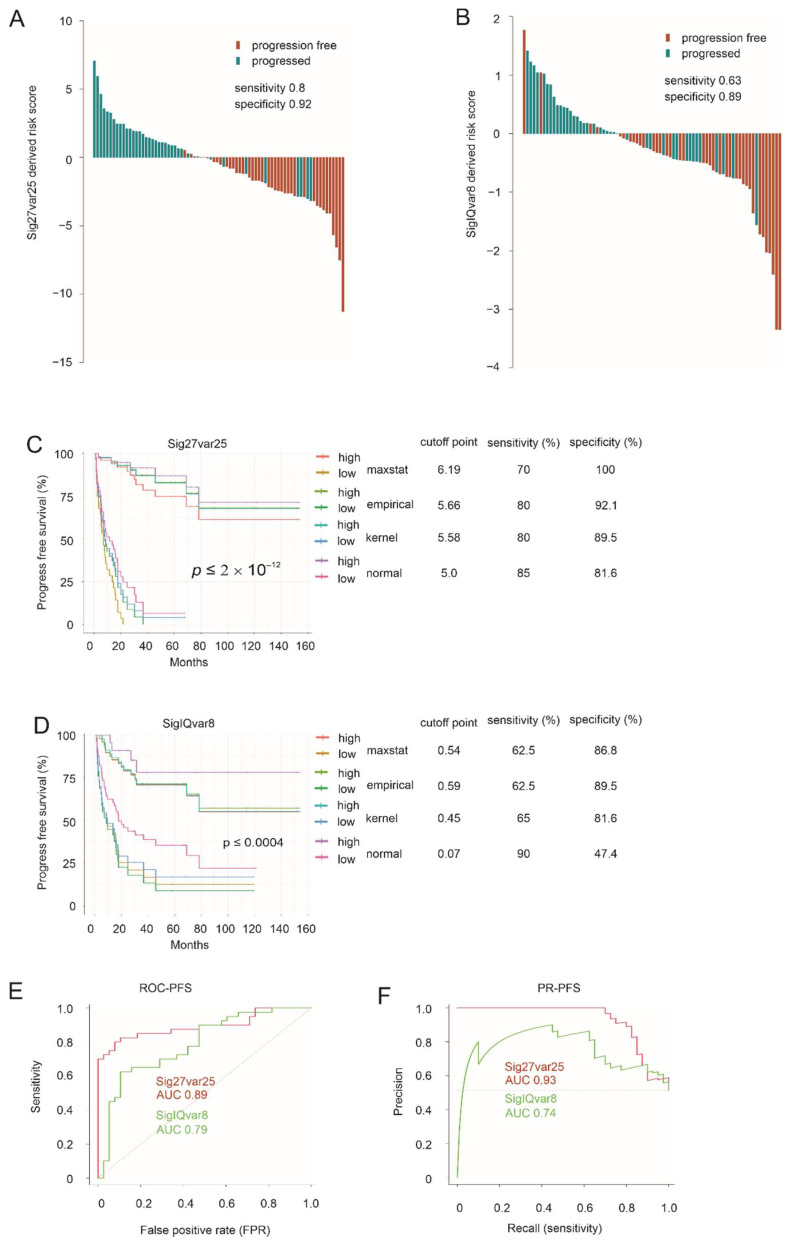
Estimation of ACC progression by Sig27var25 and SigIQvar8. (**A**,**B**) Waterfall plots for Sig27var25 (**A**) and SigIQvar8 in stratification of ACC progression risk. The progression status, sensitivity, and specificity of the risk separation are indicated. Cutoff points were estimated using the empirical methods with *n* = 1000 bootstraps and used as the baselines for waterfall plot generation using R. (**C**,**D**) Separation of ACCs with a high risk of progression from those with low risk by the indicated signatures. Methods used in cutpoint estimation, cutpoints and the respective sensitivity and specificity are indicated. (**E**,**F**) ROC and PR curves for the indicated multigene panels.

**Figure 3 cancers-14-02805-f003:**
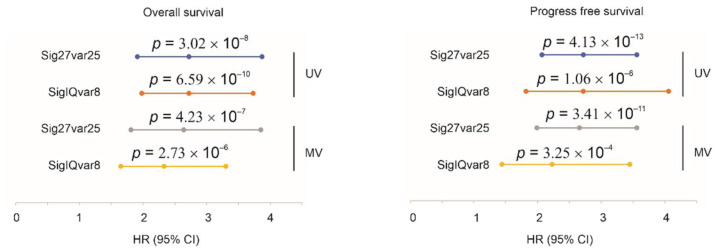
Clinical factor independent prediction of ACC progression and poor prognosis by Sig27var25 and SigIQvar8. UV: univariate Cox analysis; MV: multivariate Cox analysis. MV included age at diagnosis and stages with stages I and II were grouped into “0” and stages III and IV were classified as “1”.

**Figure 4 cancers-14-02805-f004:**
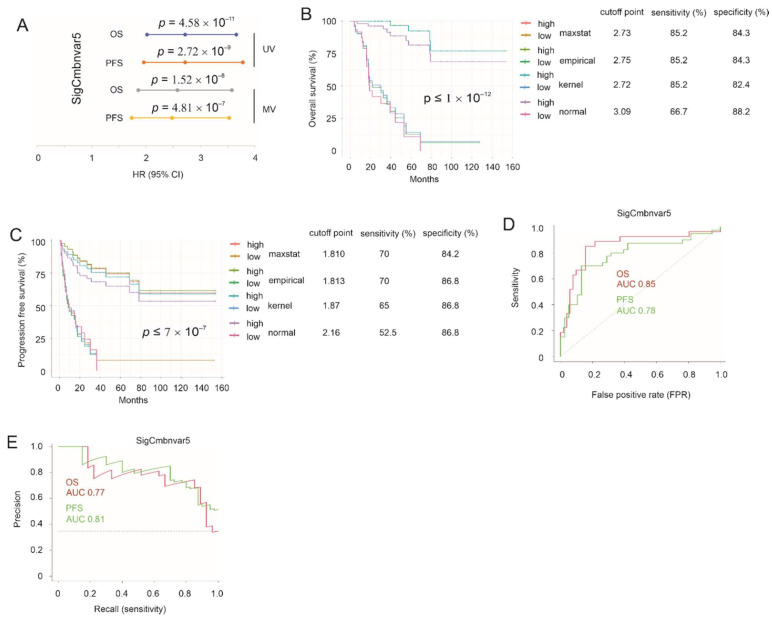
Estimation of ACC progression and fatality risks by SigCmbnvar5. (**A**) HR, 95% CI, and the respective *p*-values for SigCmbnvar5-derived prediction of OS and PFS under both univariate (UV) and multivariate (MV) settings. MV includes age at diagnosis and tumor stages. (**B**,**C**) Stratification of ACCs fatality (**B**) and progression risk (**C**) with the indicated cutoff points. (**D**,**E**) ROC (**D**) and PR curves (**E**) for discrimination of OS and PFS.

**Figure 5 cancers-14-02805-f005:**
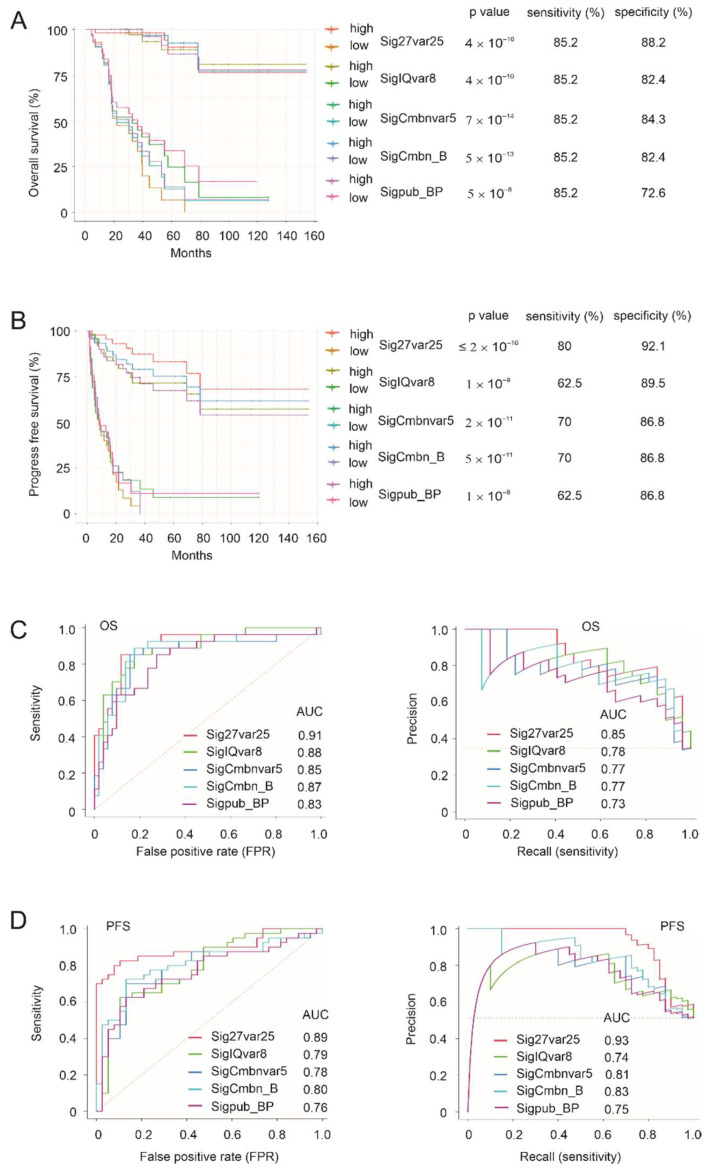
Comparison of progression and fatality risk stratification among Sig27var25, SigIQvar8, SigCmbnvar5, SigCmbn_B, and Sigpub_BP. (**A**,**B**) Cutoff points for the indicated signature scores were estimated by the empirical method. The individual survival curves, *p*-values, sensitivities, and specificities for OS and PFS are shown. (**C**,**D**) OS ROC and PR curves (**C**) and PFS ROC and PR curves (**D**) for the indicated signatures.

**Figure 6 cancers-14-02805-f006:**
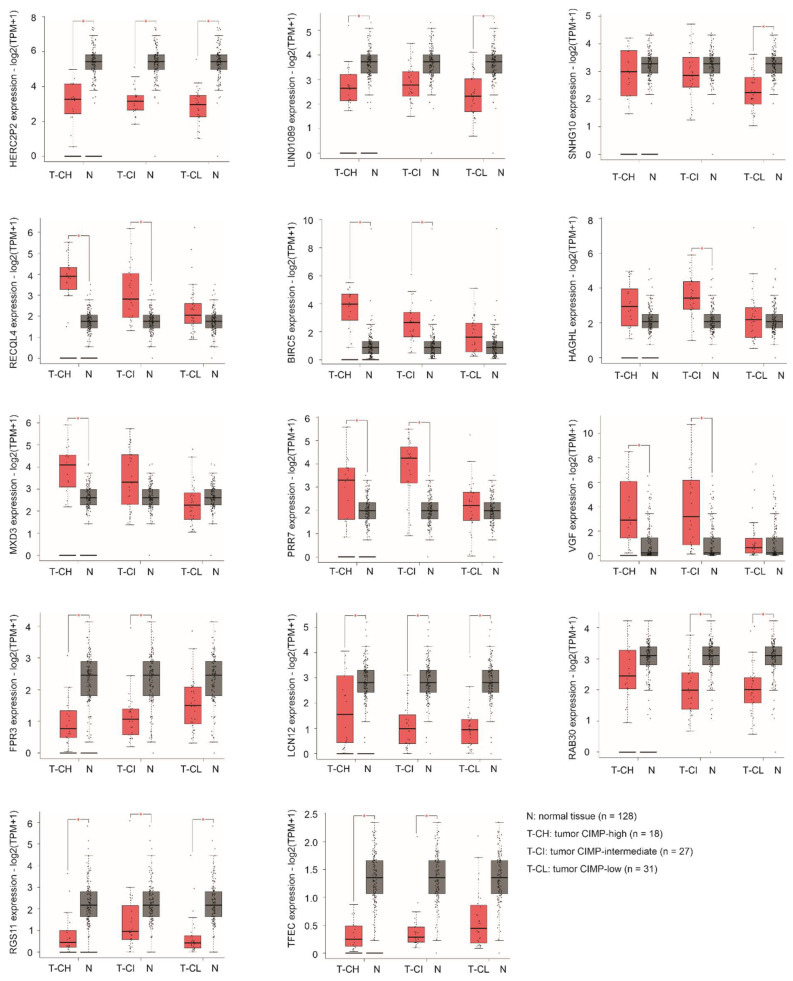
Differential expression of signature component genes. Gene expression was determined using RNA-seq data within the TCGA database organized by the GEPIA2 website [28]. * *p* < 0.05.

**Figure 7 cancers-14-02805-f007:**
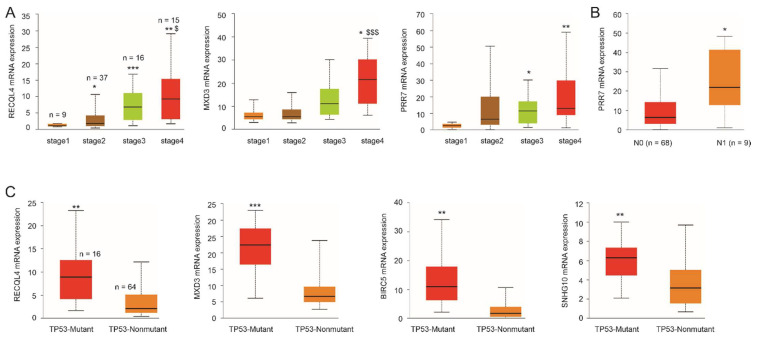
Associations of component gene expressions with tumor stage (**A**), lymph node metastasis (**B**), and TP53 mutations (**C**). Analyses were performed using the TCGA data organized by the UALCAN platform [29]. * *p* < 0.05; ** *p* < 0.01; *** *p* < 0.001 in comparison to stage 1 (**A**), N0 (**B**), and TP53-Nonmutant tumors. $ *p* < 0.05; $$$ *p* < 0.001 in comparison to stage 3 ACCs (**A**).

**Figure 8 cancers-14-02805-f008:**
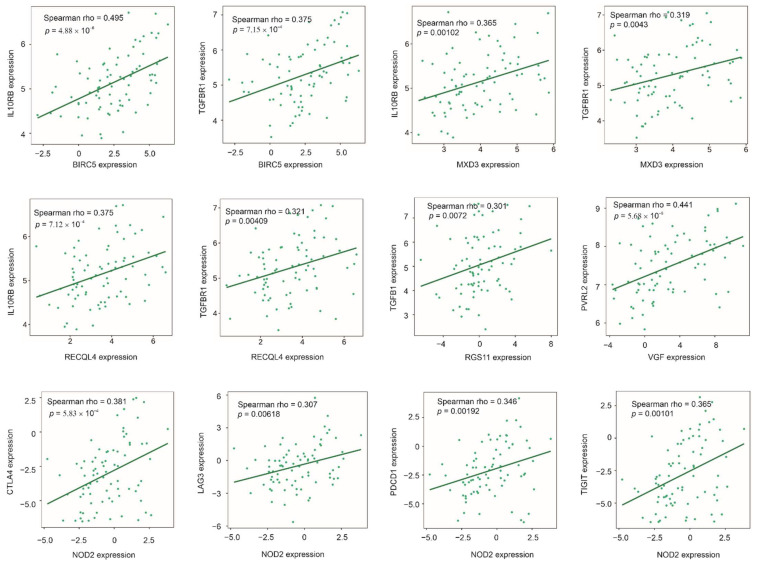
Correlations of the indicated signature component genes (*x* axis) with the indicated immunosuppressive factors (*y* axis).

**Figure 9 cancers-14-02805-f009:**
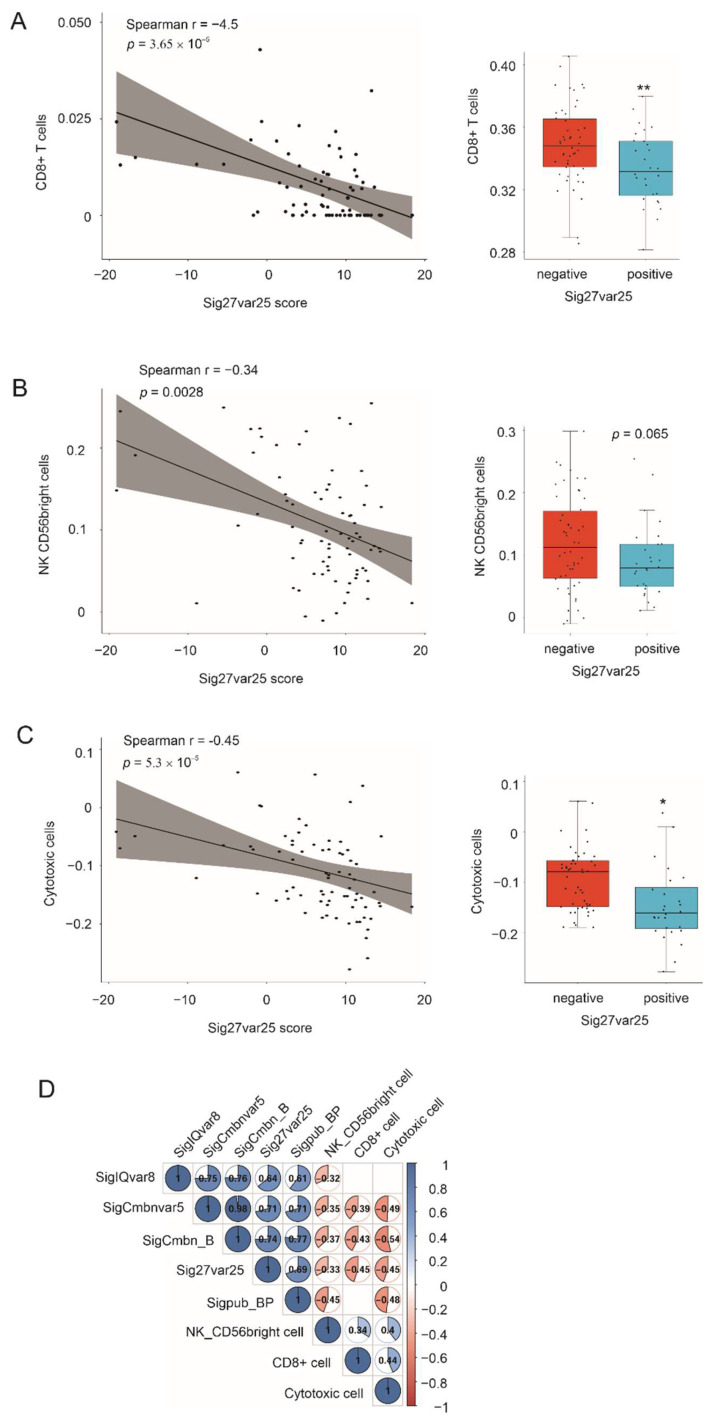
Correlations of multigene signatures with reductions of CD8+ and NK cells in ACC. (**A**) Immune cells were profiled using Xcell within the MSDIC R package, followed by the determination of the correlation of CD8+ T cells with Sig27var25 scores using the ggpubr R package (left panel). The Sig27var25 negative and positive ACC (boxplot, right panel) were defined with the cutoff point estimated using an empirical method (right panel). (**B**,**C**) NK CD56bright and cytotoxic cells in ACCs were profiled using ssGSEA within the MSDIC R package. (**D**) Scores from the indicated multigene signatures, NK CD56bright cells, CD8+ T cells, and cytotoxic cells were used to construct the Spearman correlation image with the corrplot R package. Correlations with *p* < 0.01 are included. * *p* < 0.05; ** *p* < 0.01.

**Figure 10 cancers-14-02805-f010:**
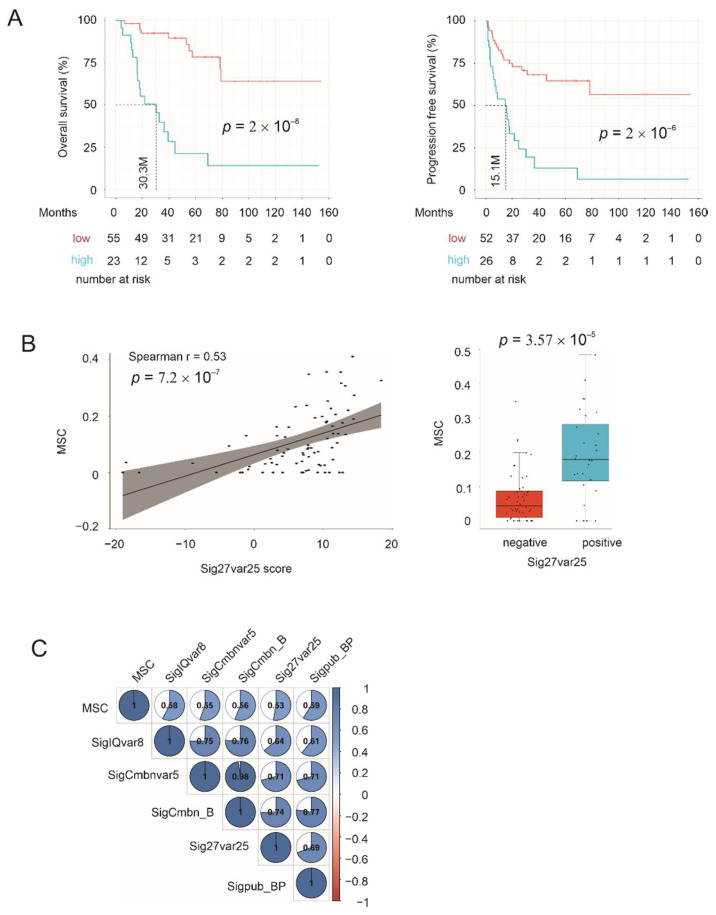
Correlation of multigene signatures with MSC. (**A**) MSCs in ACC were profiled using xCell within the MSDIC R package. Cutoff points for OS and PFS were estimated using Maximally Selected Rank Statistics, which were used to construct the survival curves. (**B**) Correlation of MSC with Sig27var25 score (left panel) and enrichment of MSCs in Sig25var25 positive ACCs (right panel). The positive and negative statuses were defined according to the empirically derived cutoff point. (**C**) Correlations of MSC with the indicated signatures were determined by Spearman correlation; all correlations are at *p* < 0.01.

**Table 1 cancers-14-02805-t001:** Univariate and multivariate Cox analyses of Sig27var25 component genes in predicting ACC poor OS and progression.

	Univariate Cox Analysis			Multivariate Cox Analysis ^1^		
Gene ^2^	HR	95% CI	*p*-Value	HR	95% CI	*p*-Value
**Poor OS**						
LINC01089	1.002	1–1.003	0.0479 *	1.002	0.9997–1.003	0.101
RGS11	1	1–1.001	0.0291 *	1	1–1.001	0.0697
MXD3	1.003	1.002–1.004	8.53 × 10^−7^ ***	1.003	1.0016–1.004	2.75 × 10^−5^ ***
BIRC5	1.003	1.002–1.004	1.45 × 10^−9^ ***	1.002	1.0014–1.003	8.79 × 10^−7^ ***
RAB30	1.008	1.003–1.013	0.00283 **	1.006	1.001–1.012	0.0122*
**worse PFS**						
LCN12	1.007	1.001–1.013	0.0187 *	1.011	1.0004–1.018	0.00172 **
VGF	1	1–1	0.00124 **	1	1–1	0.0463 *
RGS11	1.001	1–1.001	0.00271 **	1.001	1.0001–1.001	0.0207 *
MXD3	1.002	1.001–1.003	0.000676 ***	1.0017	1.0005–1.003	0.00495 **
BIRC5	1.002	1.001–1.002	6.13 × 10^−5^ ***	1.001	1.0002–1.002	0.0196 *
RAB30	1.02	1.005–1.014	1.92 × 10^−5^ ***	1.0099	1.0053–1.014	1.94 × 10^−5^ ***
NOD2	1.007	1.001–1.014	0.0237 *	1.0064	1.0003–1.012	0.0394 *
ZFHX4	1	1–1.001	0.035 *	1.001	1.0001–1.001	0.0235 *

^1^: In analysis with age at diagnosis and tumor stage (stage 1 (3 + 4) vs. stage 0 (1 + 2)); ^2^: continuous gene expression data were used in analysis; * *p* < 0.05; ** *p* < 0.01; *** *p* < 0.001.

**Table 2 cancers-14-02805-t002:** Univariate and multivariate Cox analyses of SigIQvar8 component genes in predicting ACC poor OS and progression.

	Univariate Cox Analysis			Multivariate Cox Analysis ^1^		
Gene ^2^	HR	95% CI	*p*-Value	HR	95% CI	*p*-Value
**poor OS**						
LINC01089	1.002	1–1.003	0.0479 *	1.002	0.9997–1.003	0.101
SNHG10	1.011	1.006–1.017	1.72 × 10^−5^ ***	1.01	1.004–1.016	0.000758 ***
RECQL4	1.002	1.001–1.002	9.26 × 10^−8^ ***	1.001	1.0004–1.002	0.00158 **
**worse PFS**						
LOC100128288	0.9897	0.98–0.995	0.0392 *	0.993	0.981–1.005	0.2556
SNHG10	1.006	1.002–1.01	0.00175 **	1.01	1.004–1.016	0.000758 ***
RECQL4	1.001	1.001–1.002	0.00015 ***	1.001	1.0004–1.002	0.00158 **

^1^: In analysis with age at diagnosis and tumor stage (stage 1 (3 + 4) vs stage 0 (1 + 2)); ^2^: continuous gene expression data were used in analysis; * *p* < 0.05; ** *p* < 0.01; *** *p* < 0.001.

**Table 3 cancers-14-02805-t003:** Oncogenic role of the SigIQvar8 component genes.

Gene	Oncogenic Role in ACC	Oncogenic Role in Others	Reference
LINC01089	unknown	inhibition of breast cancer metastasis	[44,45]
LOC155060	unknown	unknown	NA
LOC100128288	unknown	unknown	NA
SNHG10	unknown	promotion of resistance to tyrosine kinase inhibitors in lung cancer	[50]
RECQL4	unknown	promotion of gastric cancer	[54]
HERC2P2	unknown	suppression of cell proliferation	[51]
ATXN7L2	unknown	unclear	NA
THSD7A	unknown	unclear	NA

NA: not available.

**Table 4 cancers-14-02805-t004:** Oncogenic role of the Sig27var25 component genes.

Gene	Oncogenic Role in ACC	Oncogenic Role in Others	Reference
HAGHL	unknown	Unknow	NA
LCN12	unknown	Unknown	NA
DCST2	unknown	Unknown	NA
VGF	unknown	promotion of resistance to tyrosine kinase inhibitors in lung cancer	[56]
RGS11	unknown	a biomarker of lung cancer	[57]
PRR7	unknown	Unknown	NA
LINC01089	unknown	See Table 3	See Table 3
MXD3	unknow	promotion of medullobastoma	[58]
BIRC5	Upregulation in AC and association with ACC poor prognosis [75]	promotion of cancer progression and metastasis	[59,60]
LTC4S	unknown	a component gene of an immune signature of breast cancer	[61]
FPR3	unknown	sustain meiotic recombination checkpoint actions	[62]
RAB30	unknown	association of good prognosis in triple negative breast cancer	[63]
RIPOR2	unknown	association of immune cell infiltration and thus inhibition of cervical cancer	[64]
NOD2	unknown	immunosuppression of tumorigenesis of gastric cancer	[69]
PLXNA4	unknown	inhibition of tumor cell migration and contribution to innate immunity in working with Toll-like receptor	[65,66]
TFEC	unknown	regulation of lysosome biogenesis and mTOR activation	[74]
PI15	unknown	biomarker of cholangiocarcinoma	[70]
ZFHX4	unknown	a susceptibility locus of cutaneous basal cell carcinoma	[73]
LAMP3	unknown	a hypoxia-induced gene associated with aggressive breast cancer	[71]
HDAC9	unknown	increases in expression in bladder cancer	[72]
MCTP1	unknown	downregulation in paclitaxel-resistant ovarian cancer cells	[67]
KCNN3	unknown	suppression of bladder cancer cell migration and invasion	[68]
PCDHB8	unknown	Unknown	NA
PCDHGB2	unknown	Unknown	NA
PCDHGA5	unknown	Unknown	NA

NA: not available.

## Data Availability

The data presented in this study are available on request from the corresponding author.

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
