# Peer review of "Prediction of Adrenocortical Carcinoma Relapse and Prognosis with a Set of Novel Multigene Panels"

_cancers, 2022, doi:10.3390/cancers14112805_

Round 1

Reviewer 1 Report

The authors propose several gene panels for the prognosis and survival of adrenocortical carcinoma relapse. The methods are comprehensive with survival analysis KM plots. The manuscript has a high-quality presentation (figures/tables). The results support the hypothesis. The manuscript may gain so many interests. However, I have a few concerns:

  • The authors may discuss the clinical practice for the proposed gene panels, are they only gene expression. or may have different omics/measures, non-coding RNA, DNA Amplicon, or a mix of them.
  • The authors may discuss the findings from the survival/prognostic viewpoint. e.g. NOD2 was recently associated with the prognostic and survival of breast cancer ( I suggest highlighting PMID: 35205681). The authors may check the genes from a survival/prognostics perspective.

Author Response

We appreciate the reviewer’s positive view and insightful remarks. Here are our detailed revisions.

“The authors may discuss the clinical practice for the proposed gene panels, are they only gene expression. or may have different omics/measures, non-coding RNA, DNA Amplicon, or a mix of them.”

“The authors may discuss the findings from the survival/prognostic viewpoint. e.g. NOD2 was recently associated with the prognostic and survival of breast cancer ( I suggest highlighting PMID: 35205681). The authors may check the genes from a survival/prognostics perspective.”

Authors' response – As both comments are related; we thus address them together. Integration of multiple omics data is indeed an intriguing direction for biomarker studies and will certainly enhance risk assessment. We appreciate the reviewer for providing a recent Cancers article for a framework in integration of 3 omics to generate robust models in predicting breast cancer relapse following surgery. This system fits well with our multigene panel-derived risk scores, which we have thoroughly integrated into this revision (lines 599-615, marked with red). This addition not only adds to this manuscript but also points out an interesting research direction, for which we thank the reviewer for these remarks.

Reviewer 2 Report

The authors study the predictor potential on ACC overall survival and prognosis of panel genes initially elaborated to evaluate the recurrence of prostate cancer and clear cell renal cell carcinoma. Using this method, they demonstrate the importance of new prognosis gene that evaluate the immune response and the presence of mesenchymal stem cells in ACC. This study is very interesting and provide important information. I just have the following comments: 

-        The authors compare the prognostic efficiency of their panel versus the CpG methylation and the expression of BUB1B/PINK1 gene that have been previously published to predict ACC prognosis. However, it will also be interesting to determine if their gene panels would increase the sensitivity and specificity of the ACC classification compared to the transcriptomic approaches previously published that leads to the generation of two ACC subgroups, C1A and C1B. 

-        The authors show that their SigCmbn_B multigene panel is a better predictor than previously published Sigpub_BP but they decided to add in SigCmbn_B panel, BUB1B gene which is one of the three genes composing Sigpub_BP. The authors choose to include only BUB1B in their panel because they consider it is the only one adding information, is it right? This part should clarify 

-        The authors show the association of RECQL4, MXD3, BIRC5, and SNHG10 expression with the p53 mutated ACC. Does all the panels lead to the analysis of these genes or is this information hold better by one specific panel? Moreover, what is the opinion  of the authors on the role of this association in predicting overall survival and prognosis?  

-        The authors link the immune signature to p53 mutations so is the expression of RECQL4, MXD3, BIRC5, and SNHG10 predicting p53 status sufficient to correlate with the immune status?   

-        LINC01089 is a known regulator of Wnt/bcatenin pathway, which plays an important role in adrenocortical tumorigenesis. Moreover, as its activation has previously been associated with poor prognosis in ACC, did the authors analyze if the expression of LINC01089 is associated with the bcatenin status of the tumors? 

-       The authors demonstrate a correlation between their signature with immune escape. As hypercortisolism alters immune response through inflammation regulation, it would be interesting to know the hormonal status of their cohort and to determine if there is an association between the markers of immune status and the level of cortisol.  

Author Response

We thank the reviewer for the overall positive tone as well as the insightful comments with regard to the mechanistic and clinical aspect of this study. Here are our thorough revisions.

“The authors compare the prognostic efficiency of their panel versus the CpG methylation and the expression of BUB1B/PINK1 gene that have been previously published to predict ACC prognosis. However, it will also be interesting to determine if their gene panels would increase the sensitivity and specificity of the ACC classification compared to the transcriptomic approaches previously published that leads to the generation of two ACC subgroups, C1A and C1B.”

Authors' response – The BUB1B/PINK1 gene pair as well as C1A and C1B clustering were resulted from an elegant microarray study by de Reynies et al (Journal of Clinical Oncology 27, 1108-1115, 2009). C1A and C1B subgroups were clustered based on 746 genes and stratify ACC’s progression and prognosis risk with high levels of certainty. We have compared the sensitivity and specificity of C1A-C1B to the sensitivity and specificity of our panel Sig27var25 in predicting ACC relapse and prognosis. With the consideration in balancing sensitivity and specificity, our panel performed at least equally well; this comparison is discussed more in depth in this revision (lines 509-521, marked with red). This comparison provides an additional support for the clinical potential of Sig27var25; we thank the reviewer for this comment.

“The authors show that their SigCmbn_B multigene panel is a better predictor than previously published Sigpub_BP but they decided to add in SigCmbn_B panel, BUB1B gene which is one of the three genes composing Sigpub_BP. The authors choose to include only BUB1B in their panel because they consider it is the only one adding information, is it right? This part should clarify”

Authors' response – The exclusion of PINK1 was resulted from model selection by BeSS, which was based on its high levels of correlation with other component genes. As a result, PINK1 will not contribute to the model’s biomarker potential in predicting ACC prognosis and was thus not included in SigCmbn_B. We have added these details in this revision (lines 309-310, marked with red).

“The authors show the association of RECQL4, MXD3, BIRC5, and SNHG10 expression with the p53 mutated ACC. Does all the panels lead to the analysis of these genes or is this information hold better by one specific panel? Moreover, what is the opinion  of the authors on the role of this association in predicting overall survival and prognosis?”

Authors' response – We thank the reviewer for these insights. These 4 genes are present in all 4 multigene panels in different mixture: RECQL4 and SNHG10 in SigIQvar8; MXD3 and BIRC5 in Sig27var25; RECQL4, BIRC5, and SNHG10 in SigCmbnvar5; and RECQL4, MXD3, and SNHG10 in SigCmbn_B. Considering TP53 mutation being a driver event in aggressive ACC (C1A), their association with TP53 likely contributes to the biomarker potential of individual multigene panels. These details and our opinions are presented in this revision (lines 564-578, marked with red).

“The authors link the immune signature to p53 mutations so is the expression of RECQL4, MXD3, BIRC5, and SNHG10 predicting p53 status sufficient to correlate with the immune status?”

Authors' response – We see the interesting points raised by reviewer’s comments. The contributions of these genes to ACC-associated immune profile might be in part attributable to their relationship with TP53. We have provided our opinions in this revision (lines 578-582, marked with red).

“LINC01089 is a known regulator of Wnt/bcatenin pathway, which plays an important role in adrenocortical tumorigenesis. Moreover, as its activation has previously been associated with poor prognosis in ACC, did the authors analyze if the expression of LINC01089 is associated with the bcatenin status of the tumors?”

Authors' response – We see reviewer’s points. LINC01089 has been reported to inhibit the Wnt/β-catenin signaling in breast cancer. Its involvement in ACC remains unknown. As the reviewer has pointed out, β-catenin is a driver gene in ACC. As LINC01089 positively correlates with poor prognosis in ACC, its action is unlikely to inhibit β-catenin in ACC. This reasoning is included in this revision (lines 342-345, marked with red).

“The authors demonstrate a correlation between their signature with immune escape. As hypercortisolism alters immune response through inflammation regulation, it would be interesting to know the hormonal status of their cohort and to determine if there is an association between the markers of immune status and the level of cortisol.”

Authors' response – We thank the review for this clinical insight. Hypercortisolism is the most common presentation in ACC patients with hormone excess and the anti-inflammatory actions of corticosteroid present a clear challenge in patient selection for immunotherapy. Study of relationship between our panels and cortisol level in ACC patients is indeed interesting. Nonetheless, the TCGA ACC pan-cancer cohort does not have cortisol level data; we will thus leave this area for future research.

Reviewer 3 Report

Dear Authors,

This is an interesting article.

Here are my suggestions/comments:

  1. Title: Since this is a new panel of genes, how “effective” is actually the prediction?
  2. Title – abstract: There is a discrepancy between clinical aspects that are suggested in tile regarding survival and the actual genetic study which is very well detailed in Abstract /Results
  3. Introduction – Currently, the prognostic is rather based on Weiss score than only Ki67 proliferation index. (line 63-64)
  4. Discussion – which is the next step of clinical validation considering the multitude of aspects that should be taken into account when considering the prognostic in ACC
  5. Discursion – can you suggest a practical algorithm of genetic approach in ACC also referring/introducing your discovery?

Thank you

Author Response

We thank the reviewer for his/her positive remarks and the clinical insights.

 “Title: Since this is a new panel of genes, how “effective” is actually the prediction?”

“Title – abstract: There is a discrepancy between clinical aspects that are suggested in tile regarding survival and the actual genetic study which is very well detailed in Abstract /Results”

Authors' response – We appreciate the reviewer’s emphasis on precision. Both comments are related and will be addressed together. We have removed “effective” from the title and replaced “poor overall survival” with “prognosis” to precisely indicate the clinical aspect. We agree the revised title reflects the content of this article more closely, for which we thank the review for the above comments.

“Introduction – Currently, the prognostic is rather based on Weiss score than only Ki67 proliferation index. (line 63-64)”

Authors' response – The clinical application of Weiss score in prognosis assessment has been added (line 64).

“Discussion – which is the next step of clinical validation considering the multitude of aspects that should be taken into account when considering the prognostic in ACC

"Discursion – can you suggest a practical algorithm of genetic approach in ACC also referring/introducing your discovery?”

Authors' response – We will response to both remarks together. Our opinions on potential clinical applications of knowledge presented in this manuscript are provided (lines 534-549, marked with red). Hopefully, these opinions will open floor for discussions.